# Hybrid-Grained Dynamic Load Balanced GEMM on NUMA Architectures

**Xing Su** [†] **and Fei Lei** *,[†]

National Laboratory for Parallel and Distributed Processing, National University of Defense Technology, Changsha 410073, China; xingsu@nudt.edu.cn

* Correspondence: leifei@nudt.edu.cn

† Current address: Yanwachi Main Street 109, Changsha 410073, Hunan, China.

**Abstract:** The Basic Linear Algebra Subprograms (BLAS) is a fundamental numerical software and GEneral Matrix Multiply (GEMM) is the most important computational kernel routine in the BLAS library. On multi-core and many-core processors, the whole workload of GEMM is partitioned and scheduled to multiple threads to exploit the parallel hardware. Generally, the workload is equally partitioned among threads and all threads are expected to accomplish their work in roughly the same time. However, this is not the case on Non-Uniform Memory Access (NUMA) architectures. The NUMA effect may cause threads to run at different speeds, and the overall executing times of GEMM is determined by the slowest thread. In this paper, we propose a hybrid-grained dynamic load-balancing method to reduce the harm of the NUMA effect by allowing fast threads to steal work from slow ones. We evaluate the proposed method on Phytium 2000+, an emerging 64-core high-performance processor based on Arm's AArch64 architecture. Results show that our method reduces the synchronization overhead by 51.5% and achieves an improvement of GEMM performance by 1.9%.

**Keywords:** GEMM; BLAS; high-performance computing; linear algebra

## 1. Introduction

Dense linear algebra libraries lay the foundation for scientific and engineering computation. The Basic Linear Algebra Subprograms (BLAS) defines a collection of routines which act as basic building blocks for dense linear algebra operations. As the BLAS APIs are so widely used, processor vendors often provide BLAS implementations that are highly optimized for their processors, e.g., Intel MKL, AMD ACML and NVIDIA cuBLAS. The High-Performance Computing (HPC) community has also contributed several high-quality open-source BLAS implementations such as ATLAS [1], GotoBLAS [2], OpenBLAS [3] and BLIS [4].

The BLAS routines are categorized into three levels, level-1 for vector-vector operations, level-2 for matrix-vector operations, and level-3 for matrix-matrix operations. The three levels have different computational and memory accessing complexity. Specifically, the computational and memory accessing complexity are $O(N)$ and $O(N)$ for level-1, $O(N^2)$ and $O(N^2)$ for level-2, $O(N^3)$ and $O(N^2)$ for level-3, respectively. Among all three levels, level-3 provides the most opportunities for optimization because it performs $O(N^3)$ computation while accessing only $O(N^2)$ memory.

Among all level-3 operations, GEneral Matrix Multiply (GEMM) is of the most interest as other level-3 operations can be defined in terms of GEMM and some level-1 and level-2 operations [5]. As a consequence, the research community has spent lots of effort on optimizing GEMM for different architectures [3,6–10].

To fully exploit modern multi-core and many-core processors, GEMM is often parallelized with threading techniques such as OpenMP and pthreads. In the past decades, since the emergence of multi-core processors, the simplest strategy has been used to parallelize GEMM, in which the whole workload is equally partitioned among all threads. This simplest strategy worked well because on a homogeneous processor all threads run at roughly the same speed. So an equalized workload partition leads to a balanced utilization of processor cores. However, this is not the case on Non-Uniform Memory Access (NUMA) architectures. On NUMA architectures, the processor cores witness different memory latency when accessing different memory nodes. For GEMM, data of matrices are distributed on all memory nodes to maximize memory bandwidth and balance memory traffic. As a result, threads on different cores may run at different speeds due to the NUMA effect. The GEMM performance suffers from the variation in thread speed because the overall executing time is determined by the slowest thread.

In recent years, processor vendors have been introducing more and more cores in a single processor. To provide sufficient memory capacity and bandwidth for the processor cores, recent high-end servers and HPC nodes can have 16 or more memory chips installed on a single board. As the number of memory chips grows, more Memory Controller Units (MCU) will be harnessed to manage the memory, and future architectures will have more NUMA nodes. The equalized workload partitioning technique used in current BLAS implementations is not sufficient to achieve optimal performance on large NUMA systems.

In this paper, we present a hybrid-grained dynamic load-balancing method to reduce the penalty caused by the NUMA effect. The key idea is to allow fast threads to steal work from slow ones. Our approach is based on the work-stealing algorithm, but with several improvements specifically designed for GEMM.

The main contributions are as follows:

- We are the first to address the GEMM performance problem on NUMA architectures.
- We propose a dynamic load-balancing method to reduce the penalty of NUMA effect.
- We implemented the proposed method on Phytium 2000+, an emerging 64-core high-performance processor based on Arm's AArch64 architecture. Results show that synchronization overhead is reduced by 51.5% and GEMM performance get improved by 1.9% with our method applied.

The rest of the paper is organized as follows. Section 2 introduces the GEMM program and current parallelization techniques. Section 3 demonstrates the proposed dynamic load-balancing method. Section 4 presents and analyzes the evaluation results. Section 5 reviews the related work. Finally, Section 6 concludes.

## 2. Background

GEMM performs a matrix-multiply-accumulation operation, denoted as $C = \beta C + \alpha AB$, where $A$, $B$ and $C$ are matrices of shape $M \times K$, $K \times N$ and $M \times N$, respectively, and $\alpha$ and $\beta$ are scalars. While GEMM is algorithmically simple so that a 3-deep loop nest suffices to accomplish the computation, a high-performance implementation usually use a blocked algorithm due to the sophisticated memory hierarchies on modern processors.

Listing 1 shows the blocked algorithm for GEMM. Each loop in the original 3-deep loop nest is blocked, resulting in a total of six loops (iterated with variables $k$, $n$, $m$, $nn$, $mm$ and $kk$ in Listing 1). This blocked algorithm can be viewed as two blocking layers. The first blocking layer consists of the outer three loops and the second blocking layer is formed by the inner 3.

Figure 1 shows the structure of the blocking layers. Figure 1a shows blocking layer 1. The $n$-loop (line 5) and $m$-loop (line 8) are presented, i.e., only one iteration of the outer most $k$-loop (line 2). $A[:, k : k']$ and $B[k : k', :]$ (line 4) are referred to as $\hat{A}$ and $\hat{B}$ for the sake of brevity. In blocking layer 1, the matrices $A$, $B$ and $C$ are blocked into $M_c \times K_c$, $K_c \times N_c$ and $M_c \times N_c$ sub-matrices, denoted as $A_c$ (line 10), $B_c$ (line 7) and $C_c$ (line 11), respectively. Figure 1b shows blocking layer 2. The $nn$-loop

(line 13) and *mm*-loop (line 16) are presented and the inner most *kk*-loop (line 21) is represented by a single task (drawn in gray color). In blocking layer 2, $A_c$, $B_c$ and $C_c$ are further blocked into $M_r \times K_c$, $K_c \times N_r$ and $M_r \times N_r$ sub-matrices, denoted as $A_r$ (line 18), $B_r$ (line 15) and $C_r$ (line 19), respectively. Note that $A_c$, $B_c$ are obtained by packing $A[m : m', k : k']$ (line 10) and $B[k : k', n : n']$ (line 7) into special memory layout to guarantee continuous memory access in GEMM computation, as shown by the polylines in Figure 1b.

Listing 1: GEMM Blocked Algorithm

```
1   C = βC
2   for (int k = 0; k < K; k = k + Kc) {
3   k' = min(k + Kc, K);
4   // C = C + αA[:,k : k']B[k : k',:]  (blocking layer 1)
5   for (int n = 0; n < N; n = n + nt · Nc) {
6   n' = min(n + nt · Nc, N);
7   Bc = pack(B[k : k', n : n'])
8   for (int m = 0; m < M; m = m + Mc) {
9   m' = min(m + Mc, M);
10  Ac = pack(A[m : m', k : k'])
11  Cc = A[m : m', n : n']
12  // Cc = Cc + αAcBc  (blocking layer 2)
13  for (int nn = n; nn < n'; nn = nn + Nr) {
14  nn' = min(nn + Nr, n');
15  Br = B[:, nn : nn']
16  for (int mm = m; mm < m'; mm = mm + Mr) {
17  mm' = min(mm + Mr, m');
18  Ar = A[mm : mm',:]
19  Cr = A[mm : mm', nn : nn']
20  // Cr += αArBr
21  for (int kk = k; kk < k'; kk = kk + 1) {
22  Cr += αAr[:, kk]Br[kk,:]
23  }
24  }
25  }
26  }
27  }
28  }
```

The blocking factors $M_r$, $M_r$, $M_c$, $N_c$ and $K_c$ are carefully selected so that the sub-matrices $A_r$, $B_r$, $C_r$, $A_c$, $B_c$ and $C_c$ fit into a certain level in the memory hierarchy, with the following constraints:

$$M_r + N_r + M_r N_r \leq c_0/es/nt \tag{1}$$

$$N_r K_c + 2M_r K_c \leq c_1/es/nt \tag{2}$$

$$M_c K_c + 2N_r K_c \leq c_2/es/nt \tag{3}$$

$$N_c K_c + M_c K_c \leq c_3/es/nt \tag{4}$$

where *es* denotes the size of matrix element e.g., 8B for a double-precision floating-point number, *nt* denotes the number of threads, and $c_l$ denotes the total size of all caches on level *l*. The register file is viewed as a pseudo cache on level 0. By (1), $M_r$ and $N_r$ are so constrained that $M_r$ elements from $A_r$, $N_r$ elements from $B_r$ and $C_r$ ($M_r \times N_r$) fit into the registers (the pseudo level-0 cache). By (2), $K_c$ is so constrained that $B_r$ ($N_r \times K_c$) and two $A_r$s ($M_r \times K_c$) fit into the L1 cache. By (3), $M_c$ is so constrained that $A_c$ ($M_c \times K_c$) and two $B_r$s ($N_r \times K_c$) fit into the L2 cache. Finally, by (4), $N_c$ is so constrained that $B_c$ ($K_c \times N_c$) and $A_c$ ($M_c \times K_c$) fit into the L3 cache (if it exists).

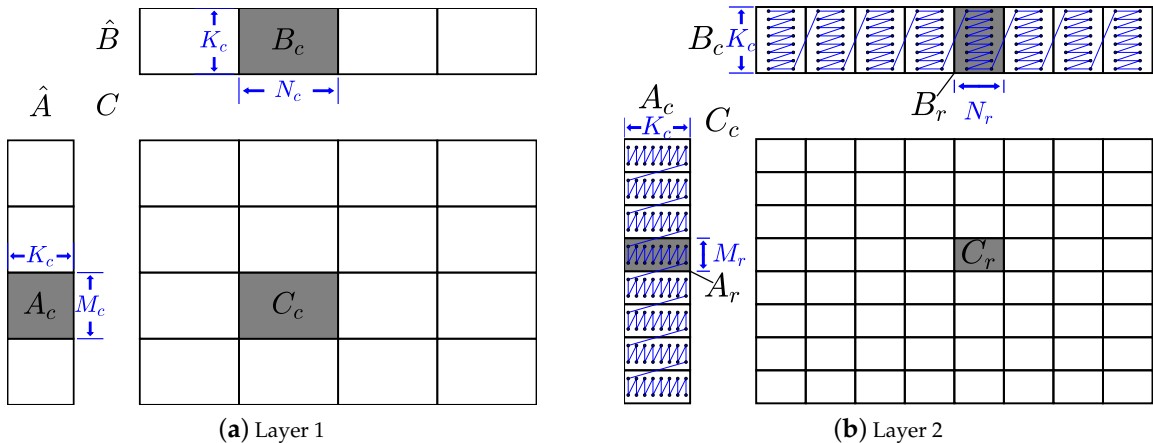

**Figure 1.** GEMM blocked algorithm. (**a**) The first blocking layer. (**b**) The second blocking layer.

The first blocking layer represents a coarse-grained workload partition and the second blocking layer represents a fine-grained one. The memory footprints of tasks on layer 1 and 2 are roughly the same as the size of L2 and L1 caches, respectively. GEMM is a computational intensive operation and Figure 1 clearly shows that workload partition on both layers have a regular shape.

In current GEMM implementations, the whole workload is parallelized on the first blocking layer, as shown in Figure 2a. There are four tasks for packing matrix $A$, four tasks for packing matrix $B$, and 16 tasks for computing matrix $C$. These tasks are statically scheduled to 4 different threads ($nt = 4$), denoted by different colors in Figure 2a. Specifically, the $i$th ($0 \leq i < 4$) thread $T_i$ gets 1 task for packing $A$ ($A_i$), 1 task for packing $B$ ($B_i$), and four tasks for computing $C$ ($C_{i,0}$–$C_{i,3}$). For thread $T_i$, the packing tasks $B_i$ and $A_i$ are first executed, Then $C_{i,i}$ is executed as its data dependencies $B_i$ and $A_i$ are resolved. Then $T_i$ checks if any other thread $T_j$ ($0 \leq j < 4$ and $j \neq i$) has finished $B_j$ because $T_i$'s computing task $C_{i,j}$ depends on $T_j$'s packing task $B_j$. If no $B_j$ is done, $T_i$ has to wait until one $B_j$ is available. This is the first kind of synchronization that may decrease GEMM performance because threads do no effective computation while waiting for other threads. There exists another kind of synchronization overhead. When $T_i$ has finished all its computing tasks $C_{i,0}$–$C_{i,3}$, before it continues to the next $k$-iteration (of the outer most loop), it must wait for all computing tasks depending on its packing task $B_i$, i.e., $C_{0,i}$–$C_{3,i}$, to be finished by their owner threads, as the buffer space of $B_i$ would be overwritten in the next $k$-iteration.

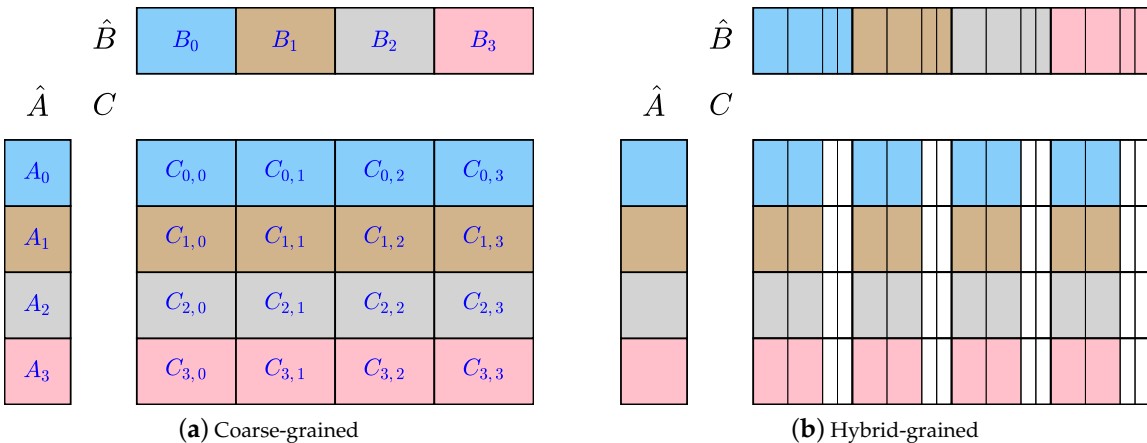

**Figure 2.** GEMM parallelization. (**a**) Coarse-grained. (**b**) Hybrid-grained.

The code for executing a single *Comp* task is encapsulated in a standalone function, known as the kernel function. Generally, the inner-most three loops in Listing 1 are factorized out as the kernel

function. The kernel function is a serial (single-thread) program which is highly optimized to achieve near-peak performance. There are plenty of research on optimizing kernel functions, which are briefly discussed in Section 5.

According to the above description of GEMM parallelization, the overall executing time of GEMM can be divided into five parts, (1) *Comp* for effective computation, (2) *Pack$_A$* for packing $A_c$s, (3) *Pack$_B$* for packing $B_c$s, (4) *Sync$_c$* and (5) *Sync$_r$* for the two kinds of synchronization overhead. The "c" and "r" subscripts are used here because the threads are waiting for data they are about to "consume" and "release", respectively. *Sync$_c$* and *Sync$_r$* harms GEMM performance on NUMA architectures. Reducing the synchronization overhead is the main motivation of this article.

## 3. Methodology

The synchronization overhead *Sync$_c$* and *Sync$_r$* occur when a thread waits for other threads to finish their tasks. A natural solution is the work-stealing algorithm, whose basic idea is to allow fast threads to steal work from slow ones. Our proposed dynamic load-balancing method is essentially a work-stealing algorithm specifically designed and optimized for the GEMM problem. Its particularities will be described in detail in this sections.

### 3.1. Hybrid Task Granularity

For the work-stealing algorithm, the choice of a proper task granularity is significant to achieve optimal performance. To reduce synchronization overhead *Sync$_c$*, the whole workload must be partitioned with a granularity smaller than that in Figure 2a. For the problem of *Sync$_r$*, we allow fast threads to steal computational tasks from slow ones. Despite the fact that threads may run at different speeds on NUMA architectures, the variation in speed should not be very large. As a consequence, a thread should take only a small piece of work each time it tries to steal, i.e., the task granularity should be quite small, even as small as a single $C_r$ computational task. Such a small granularity would dramatically increase the number of tasks, resulting in a lot of synchronization overhead. The synchronization overhead in the work-stealing algorithm includes races for locks, polling, and management of task queues. For example, with blocking factors $N_c = 512$ and $N_r = 8$ (which are typical configurations on modern multi-core processors), the fine-grained workload partition would produce 64 ($N_c/N_r$) times the number of tasks as the original coarse-grained one.

In GEMM, most of the tasks are expected to be executed by their owner threads and only a small portion of workload are done by thief threads. Based on this observation, we divide each *Comp* task in Figure 2a into two chunks, a big chunk dedicated to be executed by its owner thread, and a small chunk which can be stolen by other threads. We refer to the two chunks as the static chunk (*Chunk$_s$*) and dynamic chunk (*Chunk$_d$*), respectively. Both chunks are further divided into multiple tasks, named static tasks and dynamic tasks, correspondingly. Different granularities are selected for the two chunks, resulting in a hybrid workload partition, as shown in Figure 2b. *Chunk$_s$* and *chunk$_d$* are drawn in colored and non-colored styles, respectively. For *Chunk$_s$*, a relatively bigger granularity is selected compared to *Chunk$_d$*. As in the coarse-grained workload partition, all tasks on the *i*th row are scheduled to thread $T_i$ before execution. The difference between static tasks and dynamic tasks is that dynamic tasks are raced by threads at runtime while static tasks can only be executed by the owner threads. The granularities can be configured and tuned, which will be further discussed in Section 4.3.

### 3.2. Low-Overhead Task Management

General work-stealing algorithms use queues to manage tasks, directed-acyclic-graphs (DAG) to track dependencies between tasks, and locks to protect shared data from simultaneous access. As the GEMM program utilizes the on-chip caches so extensively that almost all data caches are occupied by matrix data, any other frequently accessed data structure may pollute the caches and affect GEMM's performance. By taking advantages of the regular structure of the GEMM program,

we design an extremely low-overhead task management mechanism which completely avoids the use of queues, trees and locks.

In Figure 2b, the number of $Pack_A$ tasks is $N_A = M/M_c$. Let $n_s$ and $n_d$ be the number of tasks in each $Chunk_s$ and $Chunk_d$, then the number of $Pack_B$ tasks is $N_B = (n_s + n_d)N/N_c$. We use a flag matrix $F_C$ ($N_A \times N_B$) to track the status of all *Comp* tasks. During GEMM execution, the valid value range for $F_C$ elements is $[0, K/K_c]$. More specifically, during the $k$th ($0 \leq k < K/K_c$) iteration of the outer most $k$-loop, $F_C(i, j)$ ($0 \leq i < N_A$, $0 \leq j < N_B$) must be either $k$ or $k + 1$. When a thread tries to acquire the dynamic task $C_{i,j}$, it performs an atomic operation `atomic_compare_and_exchange(`$F_C(i,j)$`, k, k+1)`, which succeeds if $F_C(i, j) = k$ and fails otherwise. If the operation succeeds, the thread grabs task $C_{i,j}$, leaving $F_C(i, j) = k + 1$ so $C_{i,j}$ cannot be acquired once again. An operation failure indicates that the task $C_{i,j}$ has already been acquired by some other thread. We use atomic operations instead of locks because they are more light-weighted and incurs far less overhead. For static tasks $C_{i,j}$, the owner thread access $F_C(i, j)$ with non-atomic operations because other threads never write to $F_C(i, j)$. On general architectures, the `uint8_t` type can be used as the element type of $F_C$ to support atomic operations. Besides $F_C$, two other flag matrices, $F_A$ ($N_A \times 1$) and $F_B$ ($1 \times N_B$) are used to track the status of $Pack_A$ and $Pack_B$ tasks, respectively. Unlike $F_C$, the elements of $F_A$ and $F_B$ are pointers. The pointers contain the addresses of the packed matrices on which the *Comp* tasks depend. A `NULL` pointer means that the packed data is not ready yet.

By using the flag matrices $F_C$, $F_A$ and $F_B$, we avoid explicit queues to manage tasks and DAGs to track task dependencies. By using atomic operations, no locks are needed in races for dynamic tasks. For each *Comp* task, only several bytes of the flag matrices are accessed, so the cache pollution is generally ignorable. On 64-bit processors, the memory footprints of the flag matrices can be estimated by Equation (5). With typical blocking factors $M_c = 256$, $N_c = 512$, $K_c = 256$ and configuration $n_s = n_d = 2$, an $M = N = K = 6K$ GEMM instance only requires $\Sigma = 1728$ bytes of memory, which is negligible compared to the *A*, *B* and *C* matrices.

$$\Sigma = \Sigma_A + \Sigma_B + \Sigma_C = 8\frac{M}{M_c} + 8\frac{N}{N_c}(n_d + n_s) + \frac{M}{M_c}\frac{N}{N_c}(n_d + n_s) \tag{5}$$

### 3.3. Locality-Aware Work-Stealing

GEMM is a computational intensive operation, and the optimal performance can only be achieved by exploiting on-chip caches to the maximal extent. As described in Section 2, the $A_c$ and $B_r$ sub-matrices are expected to reside in L2 and L1 caches during all iterations of the *nn*-loop and *mm*-loop, respectively. Keeping this in mind, we apply a GEMM specific optimization, the Limited Task Set (LTS) optimization, to the work-stealing algorithm,.

In a classical work-stealing algorithm, any thread $T_i$ ($0 \leq i < nt$) is allowed to race for any dynamic task, that is, $T_i$'s dynamic task set $\phi_i$ contains all dynamic tasks. The LTS optimization puts an extra limit to $\phi_i$ that $\phi_i$ only contains *Comp* tasks that depends on $T_i$'s $Pack_A$ or $Pack_B$ tasks. Figure 3 shows the distribution of dynamic task sets. Dynamic task sets of different threads are shaded by different patterns. Specifically, $T_i$'s dynamic task set $\phi_i$ spans along the $i$th row panel (*Comp* tasks that depends on $T_i$'s $Pack_A$ tasks) and the $i$th column panel (*Comp* tasks that depends on $T_i$'s $Pack_B$ tasks) of the *C* task chunks.

The LTS optimization brings several benefits:

- Only *Comp* tasks can be stolen, and all $Pack_A$ and $Pack_B$ tasks must be executed by their owner threads. So the packed matrices $A_c$, $B_c$, $A_r$ and $B_r$ live in their designated caches as in the original coarse-grained implementation.
- A thread only races for *Comp* tasks which depend on matrix data packed by itself. So the stolen *Comp* tasks are expected to run fast as the data is already (partially) in $T_i$'s local caches. This also avoids pollution to local caches.
- Any dynamic *Comp* tasks on the $C_{i,j}$ is raced by only two threads $T_i$ and $T_j$ instead of by all, thus reducing a lot of competing atomic operations.

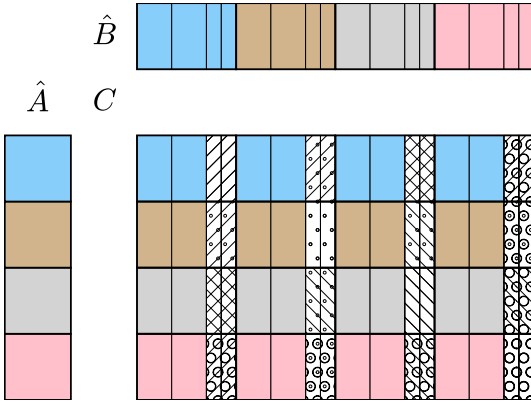

**Figure 3.** The LTS optimization.

## 4. Results

We implemented our method in the OpenBLAS [3] library and evaluated it on Phytium 2000+, an emerging high-performance many-core processor based on Arm's AArch64 architecture. We restrict our evaluation to DGEMM, as in prior work [10–12], for two reasons. First, the basic idea of the hybrid-grained load-balancing method applies to other variants of GEMM such as SGEMM, CGEMM and ZGEMM. Second, the LINPACK benchmark, which is used to build the TOP500 [13] list of world's most powerful supercomputers, relies on the DGEMM variant.

This section presents the evaluation results. First, we introduce the hardware and software environment used in our experiments. Then, we present the performance results and quantitative analysis. Finally, we give a brief discussion on tuning the granularity.

### 4.1. Environment

The Phytium 2000+ processor has 64 cores, organized into 16 core clusters with each cluster containing four cores. Figure 4a shows the memory hierarchy of the core cluster. Every core has its own L1 data cache and an L2 unified cache is shared by all cores in the cluster. Figure 4b shows the structure of the Phytium2000+ machine. L2 caches of all 16 clusters are connected by a hardware coherence network. The main memory is organized into 8 NUMA nodes, with each NUMA node hosting two clusters. Table 1 lists the hardware features of the Phytium 2000+ machine and the software environment used in evaluation. In our evaluation, we parallelize the GEMM test process with 64 threads (one thread per core).

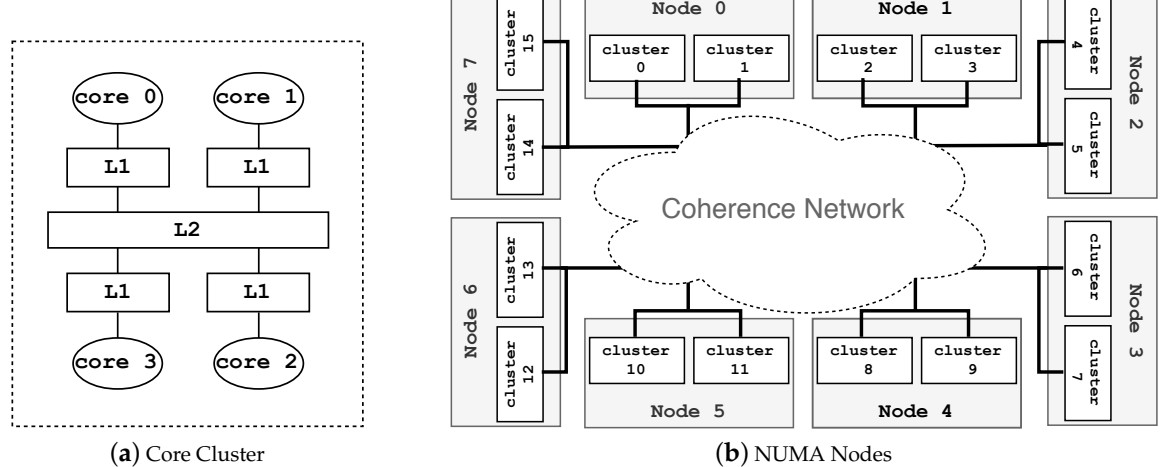

(**a**) Core Cluster                                                 (**b**) NUMA Nodes

**Figure 4.** The Phytium 2000+ processor. (**a**) Structure of core cluster. (**b**) Structure of NUMA nodes.

**Table 1.** Hardware/Software Environment.

|          | Feature          | Description                                              |
|----------|------------------|---------------------------------------------------------|
| Hardware | Architecture     | AArch64 (Arm64)                                         |
|          | Number of Cores  | 64, no hyper-threading support                          |
|          | Frequency        | 2000 MHZ                                                |
|          | SIMD             | AArch64 Neon instructions (128-bit)                    |
|          | Register File    | 32 128-bit vector registers                            |
|          | L1 Data Cache    | 32 KB, 2-way set associative, 64B cache line, LRU      |
|          | L2 Unified Cache | 2 M, 16-way set associative, 64B cache line, pseudo-random |
|          | Memory           | 128 GB, 16 GB per NUMA node                             |
| Software | Operating System | GNU/Linux 4.4.0 AArch64                                |
|          | Compiler         | GNU/GCC 6.3.0                                           |
|          | Thread Model     | OpenMP 4.0 (64 threads, one thread per core)           |
|          | BLAS             | OpenBLAS 0.3.0-dev                                      |

*4.2. Performance*

The hybrid-grained dynamic load-balancing method can be configured by three parameters, $n_s$ (number of tasks per $Chunk_s$), $n_d$ (number of tasks per $Chunk_d$) and $g$ (granularity of dynamic task). For convenience, we take the ratio of one dynamic task to the whole task chunk (sum of $Chunk_s$ and $Chunk_d$) as the granularity $g$. For instance, the configuration $(n_s, n_d, g) = (2, 2, 0.1)$ means that each task chunk is divided into two static tasks and two dynamic tasks, with each static task accounting for 40% and each dynamic tasks accounting for 10% of the whole task chunk.

We set $n_s \in [1, 2]$, $n_d \in [1, 2]$ and $g = 0.1$, and evaluate all parameter compositions and compare the performance with the coarse-grained implementation.

Both the hybrid-grained and the coarse-grained implementations use the same kernel function generated by the POCA [14] optimizer. So the only difference lies in the way they partition and schedule the computational tasks, as shown in Figure 2.

Figure 5 presents the results. For comparison, we also evaluate the performance of BLIS [4] GEMM and ATLAS [1] GEMM. The average thread efficiency, $E_{avg}$, is used to describe the performance results. The average thread efficiency is a normalized metric derived from $flops$ (floating-point operations per second), computed as $E_{avg} = flops/(nt \times \widehat{flops})$, where $\widehat{flops}$ stands for the theoretical performance peak of a single core.

All reported results in Figure 5 are obtained by running the test program five times and computing the mean value of all measured results. On each point there is an error bar representing the variation of the five measured results on that point.

ATLAS shows the worst performance among all because it use an auto-tuning methodology which has no knowledge of the underlying architecture. BLIS performs better than ATLAS, but falls behind all hybrid-grained configurations and the coarse-grained version. The reason is that BLIS's kernel (the inner-most loop in Listing 1) is not as optimal as that of OpenBLAS. For configuration $(1, 1, 0.1)$, the hybrid-grained implementation outperforms the coarse-grained implementation at most matrix sizes except for 3328, 3584, 5120 and 5888. All other hybrid-grained configurations show better performance than the coarse-grained implementation consistently at all matrix sizes. The average performance win over the coarse-grained implementation are 1.29% for $(1, 1, 0.1)$, 1.88% for $(1, 2, 0.1)$, 1.94% for $(2, 1, 0.1)$ and 1.91% for $(2, 2, 0.1)$. The results clearly demonstrate the effectiveness of our hybrid-grained dynamic load-balancing method. $(1, 1, 0.1)$ falls behind the coarse-grained implementation at some points because the number of tasks is too few ($n_s + n_d = 2$) to balance the variation in thread speed. Though it seems marginal at first glance, the 1%–2% performance improvement still makes sense for GEMM because it is so fundamental in the domain of scientific computation.

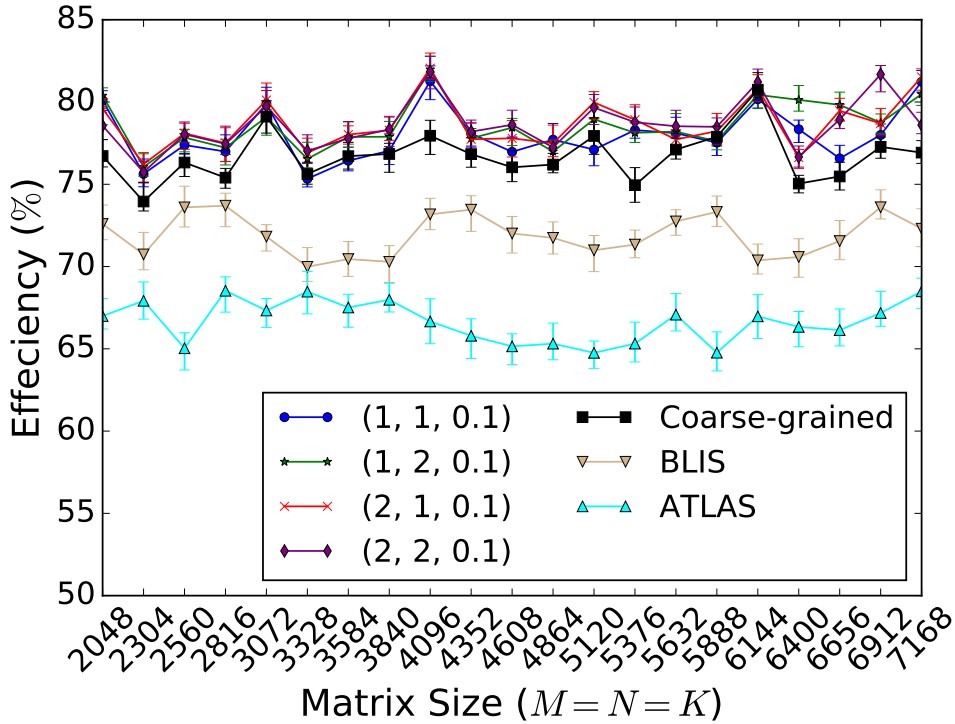

**Figure 5.** GEMM performance (64 threads). Hybrid-grained implementations are labeled with $(n_s, n_d, g)$.

To analyze the performance gain quantitatively, we measured the $Sync_c$ and $Sync_r$ overhead of the GEMM program. Figure 6 compares the synchronization overhead of the coarse-grained implementation and the hybrid-grained configuration $(2, 2, 0.1)$. The y-axis represents the ratio of synchronization overhead to the overall GEMM execution time. Both $Sync_c$ and $Sync_r$ are reduced by our hybrid-grained dynamic load-balancing method. The average overhead decreases from 4.19% to 2.03%, achieving an reduction of 51.5%. Theoretically, the improvement of GEMM performance should be $4.19\% - 2.03\% = 2.16\%$. The measured improvement ($\approx 1.9\%$) is quite close to this theoretical result, proving that the hybrid-grained dynamic load-balancing method incurs little overhead.

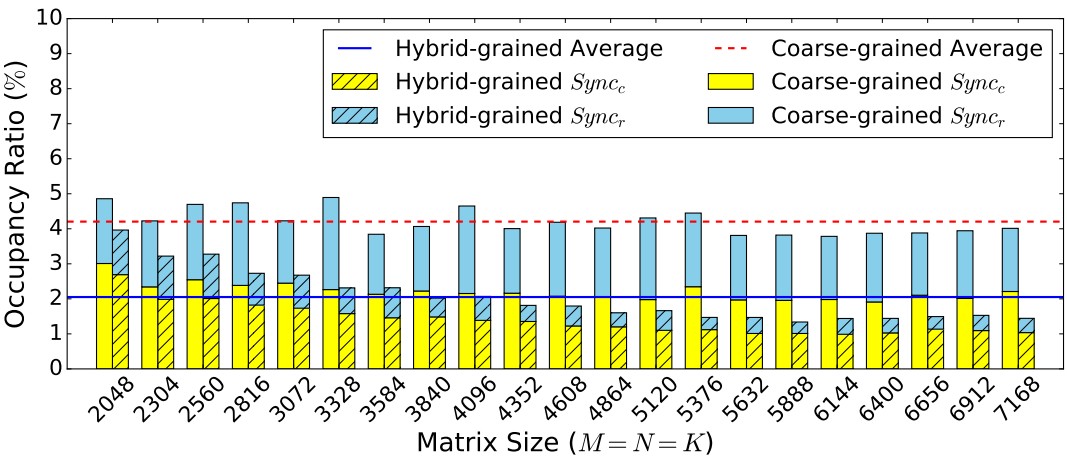

**Figure 6.** Synchronization overhead. The hybrid-grained configuration is $(2, 2, 0.1)$.

### 4.3. Tuning Granularity

In Section 4.2 configurations with $g = 0.1$ achieve good performance. To analyze how the granularity affect the overall performance of the hybrid-grained dynamic load-balancing method, we set $n_s = 2$, $n_d = 2$ and vary the granularity $g \in \{0.1, 0.2, 0.3, 0.4\}$, resulting in a total of four

configurations. Figure 7 presents the performance results of these four hybrid-grained configurations, as well as the coarse-grained implementation.

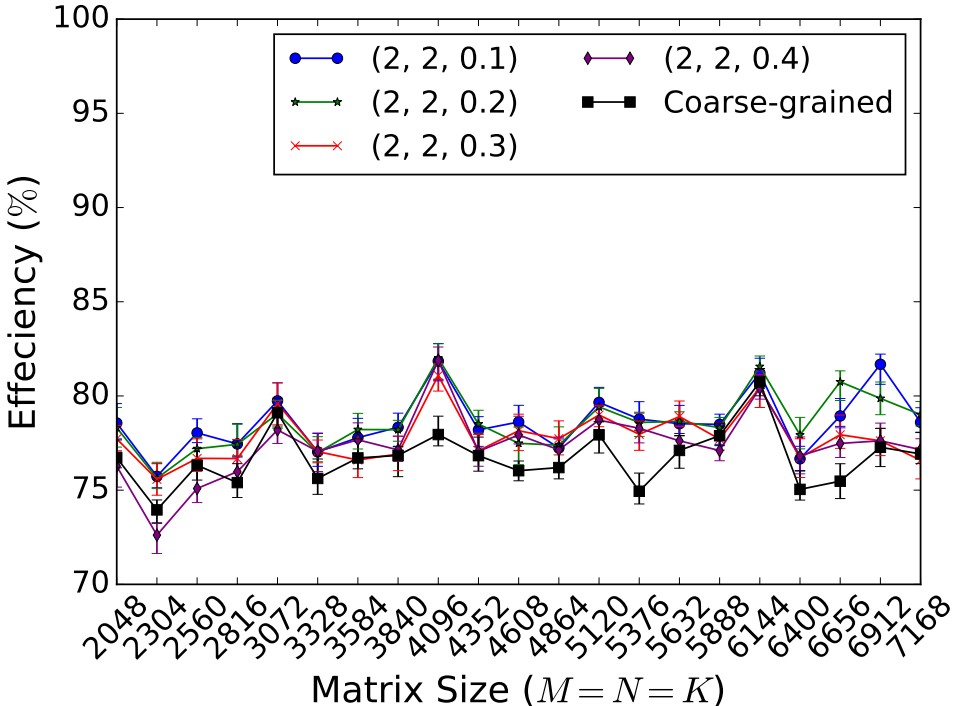

**Figure 7.** GEMM performance (64 threads).

All hybrid-grained configurations perform better than the coarse-grained implementation, though there exists a few points at which $(2, 2, 0.4)$ falls behind. The average thread efficiency of the hybrid-grained configurations are 78.68% for $(2, 2, 0.1)$, 78.60% for $(2, 2, 0.2)$, 77.80% for $(2, 2, 0.3)$ and 77.42% for $(2, 2, 0.4)$. The law is that smaller granularity shows better performance. The results confirm the design in Section 3.1. For configurations $(2, 2, 0.3)$ and $(2, 2, 0.4)$, the dynamic tasks are larger than the static ones, which violates the design illustrated in Section 3.1. So they show suboptimal performance compared to configurations $(2, 2, 0.1)$ and $(2, 2, 0.2)$. In general, $g \leq 0.2$ is a reasonable setting for the hybrid-grained dynamic load-balancing method.

## 5. Related Work

The idea that all level-3 BLAS operations can be built on top of a high-performance GEMM implementation was first proposed in [5,15]. Optimizing GEMM has always been the central task in developing dense linear algebra software since then. The GotoBLAS library [2] and its successor OpenBLAS [3], are instantiated based on this insight. Optimization of GEMM has two aspects. One is developing fast kernel functions to accomplish the computation of *Comp* tasks, and the other is workload partition and parallelization, which is the focus of this article. As far as we know, this paper is the first to address the problem of GEMM parallelization on NUMA architectures.

There are several approaches for obtaining optimized kernels, yielding different tradeoffs between performance and portability. In GotoBLAS [2], OpenBLAS [3] and BLIS [4], the kernels are usually written by domain experts in assembly. ATLAS [1] adopts the auto-tuning method to automatically generate kernels with different parameters in C and find the best-performing one by running them on the actual computing system. POET [12,16,17] and AUGEM [11] use a directive-based programming approach. POCA [14] is a compiler-based approach which generates and optimize kernels automatically and portably.

The blocking factors $M_r$, $N_r$, $M_c$, $N_c$ and $K_c$ are essential to GEMM performance. ATLAS [1] relies on auto-tuning to determine optimal values for these factors. Analytic techniques [18–20]

can also be used instead of auto-tuning. In multi-thread contexts, the *m*-loop (line 8 in Listing 1) is generally parallelized. BLIS [10] allows developers to specify a sophisticated configuration so that any combination of the *n*-, *m*-, *nn*-, and *mm*-loops can be simultaneously parallelized to suit complex architecture features like multi-sockets and hyper-threading.

While our hybrid-grained dynamic load-balancing method is specially designed for dense linear algebra computation on shared memory parallel architectures, generic dynamic load-balancing techniques have been studied extensively for distributed computing architectures. Olga et al. [21] proposed a decoupled load-balancing algorithm to enable the load balance computation to run concurrently with the application so that the scalability of the load-balancing algorithm gets improved. Patni and Aswal [22] proposed a distributed load-balancing algorithm for grid architectures. Kaur and Sharma [23] uses a "Central Load Balancer" formula to balance the burden among virtual products with reasoning data center. In [24], a hierarchical (2 levels) dynamic load-balancing model is proposed to compromise between centralized and fully distributed load-balancing schemes. The LBPSA algorithm [25] aims at balancing the workload on multiprocessor systems in real-time contexts to reduce response time and improve resource utilization. Stavros and Manos [26] address the shortage of the general block-cyclic redistribution problem on non-all-to-all networks.

## 6. Conclusions

In this article, we present a hybrid-grained dynamic load-balancing method to reduce synchronization overhead in parallelized GEMM programs on NUMA architectures. The proposed method is essentially a work-stealing algorithm with GEMM specific optimizations. Experimental results show that this method works effectively on the Phytium 2000+ 64-core machine with eight NUMA nodes. In the future, we will consider higher level dense liner algebra computations such as LU factorization Cholesky factorization and singular value decomposition. The workload partition of these higher level computations is not as regular as GEMM, which may require interesting enhancements to our hybrid-grained dynamic load-balancing method. We would also like to apply the proposed method to real-world applications in distributed architectures like for instance, the array redistribution problem.

**Author Contributions:** X.S. conceived design and methodology; F.L. assisted in software and experiments; Both authors participated in writing and reviewing the manuscript.

**Funding:** This research was funded by the National Key Research and Development Program of China (NO. 2017YFB0202003), Innovative Team Support Program of Hunan (2017RS3047).

**Acknowledgments:** The authors would like to thank Wanqing Chi and Ruibo Wang for providing a supported Linux kernel used in our evaluation.

**Conflicts of Interest:** The authors declare no conflict of interest.

## Abbreviations

The following abbreviations are used in this manuscript:

| | |
|---|---|
| BLAS | Basic Linear Algebra Subprograms |
| GEMM | GEneral Matrix Multiply |
| NUMA | Non-Uniform Memory Access |
| HPC | High-Performance Computing |
| MCU | Memory Controller Unit |
| DAG | Directed Acyclic Graph |
| flops | FLoating-point Operations Per Second |

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
