# Peer review of "Hybrid-Grained Dynamic Load Balanced GEMM on NUMA Architectures"

_electronics, doi:10.3390/electronics7120359_

Round 1

Reviewer 1 Report

This paper presents a dynamic load balancing method, which allows fast threads to take over work assigned to slow ones, on heterogeneous platforms. The idea is interesting, but  certain issues are raised and have to be addressed. 

1) Experimental Results:

The authors state that "In our evaluation, all GEMM instances are parallelized with 64 thereads, to fully exploit the hardware capability. Does this mean that, each core can have a maximum of 64 threads (assuming that each instance can run on every core)?  If so, then i believe that the authors have not taken into account the effect of hyperthreading, which would dramatically reduce performance (too many context switches). If not, then how many threads are run per processor? What is the effect of core/thread numbers on the proposed solution?

Generally, hyperthreading is a very serious issue and should be included in the discussion following the results. 

Another issue is that the authors compare their work to a typical coarse-grained workload partition, which looks more or less like a round robin task assignment.  In the Related Work section, the authors listed some approaches to optimize GEMM. Aren't they comparable to the proposed scheme? If so, why don't the authors present comparison results? Please explain.

Finally, i think the authors need to explain the fact that the proposed scheme has lower performance compared to the coarse-grained workload partition, fow certain matrix sizes ) like 3328, 3584, 5888....) as presented in par. 4.2. Is there any explanation or it is a random result? If this results appear randomly, how can they be improved? It is about 25% of the matrix sizes examined that the proposed scheme does not have better efficiency. In my opinion this percentage is high. Also, what will happen if we keep increasing the matrix size? Please answer CLEARLY the same questions for Figure 7.

2) Related Work

In my opinion, the related work includes rather old papers.Only 4 out of 20 papers presented there are after 2015, which concerns me. Even if we assume that is not a large number of papers dealing with GEMM optimization, the subject of Dynamic Load Balancing in multiprocessor systems has been considered in many papers. At least, the authors should present some of them, just for completeness of the References Section, which needs updating. 

Examples

[1] Jagdish Chandra Patni ; Mahendra Singh Aswal, "Distributed load balancing model for grid computing environment", 2015 1st International Conference on Next Generation Computing Technologies (NGCT). 2015, pp 123-126.

[2] Simranjit Kaur ; Tejinder Sharma, "Efficient load balancing using improved central load balancing technique", 2nd International Conference on Inventive Systems and Control (ICISC), pp 1-5, 2018.

[3] Lipika Datta, "A new task scheduling method for 2 level load balancing in homogeneous distributed system",  International Conference on Electrical, Electronics, and Optimization Techniques (ICEEOT), pp 4320 - 4325, 2016.

[4] Divya Jain ; Sushil Chandra Jain, "Load balancing real-time periodic task scheduling algorithm for multiprocessor enviornment 2015 International Conference on Circuits, Power and Computing Technologies [ICCPCT-2015]", pp: 1-5, 2015.

[5] Stavros Souravlas and Manos Roumeliotis, "Scheduling array redistribution with virtual channel

support", Journal of Supercomputing, Journal of Supercomputing, Springer, Volume 71, Issue

11, pp. 4215-4234, November 2015.

[6]  Olga PearceTodd GamblinBronis R. de SupinskiMartin SchulzNancy M. Amato, "Decoupled Load Balancing", Proceedings of the 20th ACM SIGPLAN Symposium on Principles and Practice of Parallel Programming, 2015.

These are just examples of works that try to balance the workload in parallel/distributed applications. There are many more, which can be added, so that an interested reader can understand the general framework of the load balancing problem and grasp a better view on where his work is exactly placed within this framework. 

Also (this is not obligatory but it could be helpful to improve the quality of this work) , the authors could also apply their work in a real application that uses arrays. An example is the array redistribution problem (see [5] of the above references), which is a typical problem where the load is distributed in a round -robin fashion (quite similar to the coarse-grained distribution). A real application, would add extra value to this work. 

3) English Language.

I have seen a number of typos, that need to be corrected. I only give two examples here, but there are more (perhaps a native speaker could help in making the paper more presentable)

a) p.4 (bottom): For the work-stealing algorithm, the choose of a proper ->

Replace choose with choice

b) p. 7 Results: We implemente -> replace impelmente with implemented

There are many typos like these, please correct. 

4) Conlusions

Please provide future work that needs to be done. At least, the implementation of the proposed scheme on real applications can be referred here, with a few hints on how it can be done. Please refer to the problem in [5], it can be easily applied using your work. 

Overall, there is a good idea in this work. I would like to see a revised version, that addresses ALL the issues raised. 

Author Response

The uploaded file is a combination of two parts.

1. The point-to-point response (page 1-7)

1. The revised manuscript (page 8-20)

We upload both parts because in the point-to-point response we refer to line numbers in the revised manuscript.

Reviewer 2 Report

The authors introduce an interesting idea applied to an important kernel for dense matrix libraries such as BLAS.
The authors compared to a coarse-grained implementation and even though they have mentioned in the related work several works that do span from fine-tuning to compiler techniques, it would have been great to compare against some of those techniques.
Even though the overheads are significantly reduced but it seems the effect of that on the performance was limited below 2% gains which suggests that the work-stealing overhead is none trivial and is being offset.

I would encourage the authors to run experiments on their testbed of the other states of the art techniques that they pointed out in their related work section.

Author Response

The uploaded file is a combination of two parts. 1. The point-to-point response (page 1-2) 1. The revised manuscript (page 3-15) We upload both parts because in the point-to-point response we refer to line numbers in the revised manuscript.

Round 2

Reviewer 1 Report

The authors have addressed my major issues in a very satisfactory manner. Specifically:

1) They addressed my very important concern regarding hyperthreading by explaining that there is one thread per core.

2) They provided some comparable results of their scheme against BLIS and ATLAS . These results have also been explained. 

3) They provided more recent related papers as requested, so that their Related Work section is more updated and complete.

4) They made a good number of corrections as far as the use of the English language is concerned.

Thus, i have to agree that this paper can now be published as it is.

Author Response

Thanks for your revision.

Reviewer 2 Report

The following is the response of the authors to an issue that was raised during the first round of reviews:

===========================

We also mentioned research projects POET and AUGEM. Their research target is to improve the performance of single-thread GEMM by applying compiler techniques such as instruction scheduling and loop tiling. So the comparison to these projects is not meaningful in this paper. Furthermore, POET and AUGEM only work on the x86 architecture. As a result, we cannot evaluate them on the AArch64 Phytium 2000+ processor. Finally, we mention the POCA project, which was previously proposed by our research group. Like POET and AUGEM, POCA is also a compiler method aiming to improve single-thread GEMM performance.

==========================================

I do agree that an x86 compiler is not fair to ask to compare against.

Yet, comparing against a compiler technique is a fair request in my opinion especially that the authors have their own infrastructure that did use a compiler.

I would even say that the technique is not unique to an Arm processor thus, even porting the technique to x86 and comparing against the compilers implemented there is not too hard to do.
I think the authors can easily compare against their own compiler work with minimal effort.

The authors run on a real machine? Right? Not on a simulator, right?

Normally there is variability in the results obtained if you run the application several times.

I am afraid with gain below 2%, if you show the min, max and mean using uncertainty bars around your results, the results might show no gains or even degradation in performance.
Can you run each experiment 10 to 20 times and report statistics on the observed results?
